# Intra-operative application of ultra-high frequency ultrasound facilitates differentiation of bowel wall characteristics between ganglionic and aganglionic segments during transanal endorectal pull-through for Hirschsprung disease
Christina Granéli [1,2] ✉, Maria Evertsson[3], Tobias Erlöv[4], Tebin Hawez[1], Kristine Hagelsteen [1,2], Louise Tofft[1,2], Tomas Jansson [3,5], Magnus Cinthio[4] & Pernilla Stenström[1,2]

## Abstract

**Background** The hypothesis was that ultra-high-frequency (UHF) ultrasound can distinguish ganglionic from aganglionic bowel during surgery for Hirschsprung disease. The aim was to assess UHF ultrasound for differentiating ganglionosis from aganglionosis in children undergoing rectosigmoid Hirschsprung disease surgery.
**Methods** The child's intestine was examined intra-operatively using the Vevo MD (Fujifilm VisualSonics, Toronto, ON, Canada) system equipped with the UHF70 transducer; 50 MHz in center frequency. Collected images were analyzed and results presented as median.
**Results** Twenty-one patients were examined intra-operatively with UHF ultrasound. The muscularis interna was thicker in ganglionic bowel compared to aganglionic: 0.298 vs 0.599 ($p < 0.001$), and the ratio of the muscularis interna/muscularis externa was greater; 0.621 vs 1.225 ($p < 0.001$). The echogenicity was higher, i.e., whiter, in the aganglionic submucosa 104.5 vs 81.6 ($p < 0.016$).
**Conclusion** The use of intra-operative UHF ultrasound shows great promise in the determination of ganglionic versus aganglionic bowel.

## Plain language summary

Hirschsprung disease occurs when part of the intestine lacks nerve cells, causing severe constipation in children and requiring surgery. During surgery, surgeons must determine where healthy bowel begins, which currently involves biopsies that are time-consuming and sometimes imprecise. In the first study of its kind, we tested whether ultra-high frequency ultrasound could help. Using a high-resolution probe during operations, we scanned the intestinal wall in 21 children. We found clear, reliable differences: healthy bowel had thicker muscle layers and a higher layer ratio than diseased bowel. This technique could provide surgeons with real-time guidance, reduce the need for biopsies, and make surgery safer and faster for children with Hirschsprung disease.

---

Hirschsprung disease is one of the most common causes of neonatal bowel obstruction and affects approximately 1:5000 live births[1]. The obstruction is caused by a lack of ganglion cells in the submucosal and myenteric nerve plexus of the intestinal wall, which leads to a constriction of the affected aganglionic intestine[2]. The affected segment starts in the rectum, extending in the oral direction, most commonly involving the recto-sigmoidal part of the bowel, but the exact length varies individually. To establish the extent of the ganglionosis in the bowel aimed for the colorectal anastomosis, i.e., to ensure that no aganglionic bowel is left, fresh frozen biopsies are obtained intra-operatively[3,4]. The need for intra-operative biopsies means that the child must endure a prolonged time under anesthesia while the samples are under histopathological examination.

---

[1]Department of Clinical Sciences Lund/Pediatrics, Lund University, Lund, Sweden. [2]Department of Pediatric Surgery, Children's Hospital, Skåne University Hospital, Lund, Sweden. [3]Department of Clinical Sciences Lund/Biomedical Engineering, Lund University, Lund, Sweden. [4]Department of Biomedical Engineering, The Faculty of Engineering, Lund University, Lund, Sweden. [5]Digitalisering IT/MT, Skåne Regional Council, Lund, Sweden. ✉e-mail: christina.graneli@med.lu.se

To avoid extra waiting time, in the case of need of repeated frozen biopsy, the surgeon might opt to collect the frozen biopsy from a more proximal site than needed, which might result in a longer part of the intestine being removed than necessary. A method to determine more accurately where to collect the biopsy from, or even to delineate the ganglionic bowel, is therefore warranted.

The introduction of ultrasound in the field of medicine has led to various improvements in the diagnostic and treatment options for patients. In recent years, ultra-high frequency ultrasound (UHF ultrasound) transducers have been developed that operate at much higher frequencies than the transducers commonly used clinically, which transmit frequencies of 2–18 MHz, capturing tissue depths of 2–15 cm. Instead, the UHF-ultrasound transducers transmit frequencies up to 70 MHz, which results in a higher resolution and more detailed view of the examined tissue at the cost of image depth[5]. UHF ultrasound has been shown great promise in, for example, diagnostics of different types of skin anomalies[6–8]. Within the field of pediatrics there are still very few reports regarding the use of UHF ultrasound. One report has emerged using the technique to evaluate gastrointestinal changes in neonatal patients[9], and in a previous study of ours, UHF ultrasound showed promising results in aiding in distinguishing aganglionic from ganglionic bowel wall ex vivo[10]. Supporting the latter, histoanatomy of bowel wall as depicted on UHF ultrasound images, has been shown to correlate well to histopathology[11]. Additionally, measurable histoanatomic differences between the aganglionic and ganglionic segments have been identified using UHF ultrasound on ex vivo examined bowel[12].

Yet these promising studies have only been undertaken on ex vivo samples. Therefore, to move closer to the clinical application of UHF ultrasound, it is essential to investigate its use on the bowel wall in vivo.

The aim of this study was to determine UHF ultrasound's effectiveness in distinguishing ganglionic from aganglionic bowel wall during trans-anal endo-rectal pull-through (TERPT) procedure for rectosigmoid Hirschsprung disease.

## Material and methods

### Patients

The study was conducted at a tertiary center for pediatric surgery serving a population of approximately 5 million residents with a birth rate of 50,000 live births per year. Children diagnosed with Hirschsprung disease during a period from the 1st of January 2020, to the 30st of September 2024, were eligible to participate in the study. Inclusion criteria were rectosigmoid aganglionosis operated on with TERPT and assistance of a subumbilical mini-laparotomy, age 0–12 months, and no preoperative stoma. Exclusion criteria included aganglionosis extending beyond the recto-sigmoid, lack of consent, and laparoscopic procedures. All primary diagnosis was performed by rectal suction biopsy, and all children had a contrast enema before the operation to estimate the length of the aganglionic segment preoperatively.

Included patients underwent a TERPT procedure with assistance of a subumbilical mini-laparotomy. A muscular cuff was constructed approximately 2 cm proximal to the dentate line, in accordance with a modified Soave technique. Ultra-high frequency ultrasonographic imaging was acquired from the serosal aspect of the intestine, utilizing longitudinal orientations. The procedure was done via a sub-umbilical incision to collect full wall biopsies to be freshly frozen and then examined. Intraoperative biopsies were collected approximately 5 cm proximal to the suspected transition zone. The biopsies were subsequently transferred to the pathology department for freezing, sectioning, and preparation. All specimens were evaluated by pediatric pathologists.

Background data were collected from a retrospective chart database regarding age at surgery, weight at surgery, length of resected bowel, gender, and perioperative complications.

### Equipment

The UHF ultrasound examination was undertaken with the intent to collect imaging data from both the aganglionic and ganglionic walls. The examination took place while waiting for the histopathological analysis of the fresh frozen biopsy. The timing was chosen so as not to prolong the total time under anesthesia. The UHF ultrasound scanner used was the Vevo MD with a linear array UHF70 transducer (Vevo MD Fujifilm VisualSonics, Toronto, ON, Canada) with a frequency bandwidth of 29–71 MHz, axial resolution down to 30 μm, lateral resolution down to 65 μm, and a maximal imaging depth of about 10 mm.

### UHF ultrasound examination in vivo

Sterile aquasonic® 100 ultrasound gel was applied to the transducer before it was wrapped in a sterile transducer cover. Similar gel was used on the bowel wall during the examination. Images were obtained from the anti-mesenteric side of the colonic wall, specifically over the taenia, to ensure a consistent anatomical landmark and allow all measurements to be performed as uniformly as possible. All ultrasound examinations were performed clinically by the same three pediatric surgeons. One examiner maneuvered the transducer, and another worked the control panel. Longitudinal images were saved from the most distal part of the bowel, i.e., the distal sigmoid, which was within reach for the transducer, at the suspected aganglionic bowel, as well as from the ganglionic bowel proximal to the fresh frozen biopsy site. According to the study protocol, the length from the anal verge was noted for the collected images. After the bowel was resected, the pathologist confirmed the presence or absence of ganglion cells at these specific sites. All fresh frozen biopsies were treated with formaldehyde and immunohistochemical dyes according to the local guidelines (including hemotoxin eosin, calretinin, and S100) to confirm the results from the initial intra-operative examination.

### Software for measurements and calculations

A dedicated in-house software developed in MATLAB (MathWorks Inc., Natick, MA, USA) for semi-automatic measurements for the bowel wall analyzes was used to analyze the UHF ultrasound images[13]. Longitudinal images of the bowel wall layers, scanned from the bowel's serosal surface, were analyzed. For each image, a region of interest (ROI) of 5 mm covering the best image quality and the best representation of the different layers of the bowel wall was chosen. The muscularis externa and interna were outlined manually within the ROI (Fig. 1). The mean thickness and mean echogenicity (image whiteness) in each anatomic layer were calculated automatically by the inhouse-software at intervals of 32 μm corresponding to 156 points within the ROI. Echogenicity of the intestinal layers was measured within the same ROI as the thickness. A higher amplitude corresponded to a whiter image appearance. Measurements were carried out within the muscularis externa, muscularis interna, and the submucosa. As the submucosa is the anatomical layer furthest away from the transducer (about 1 mm) when undertaking examinations from the serosa, the submucosa's inner limitation is not clearly visible in the image. Thus, a standardized predetermined thickness of 0.18 mm was used, and mean echogenicity was calculated. Ratios of the muscularis layers' thicknesses and echogenicity were calculated for each patient, and these ratios were compared between aganglionic and ganglionic intestinal wall with the patient acting as his or her own control.

To evaluate the repeatability of the measurements, two independent researchers performed the measurements: one pediatric surgeon (CG) and one engineer (ME). Each researcher chose their own ROI for each image.

### Statistics

Thicknesses (millimeters) and image echogenicity (unitless) regarding the layers in bowel wall were presented in median (range) as well as in mean (standard deviation; SD). For the relative difference, ratios of thicknesses of the muscularis interna/externa and ratios of the echogenicity of the muscularis interna/muscularis externa, muscularis interna/submucosa, and muscularis externa/submucosa were calculated. Calculations were made using Microsoft Excel Version 16.91. Statisticians at the Department of Clinical Studies, Sweden Forum South, assisted in the selection of the statistical methods, and they also performed a quality control of all analyzes. Each patient served as their own control, and the results were confirmed to

**Fig. 1 | Ultrasound images from two patients diagnosed with Hirschsprung disease.** The image on the left demonstrates aganglionic bowel segments, whereas the image on the right depicts normally ganglionated intestinal tissue. Distinct anatomical layers of the intestinal wall are delineated using separate bounding markers to facilitate structural comparison.

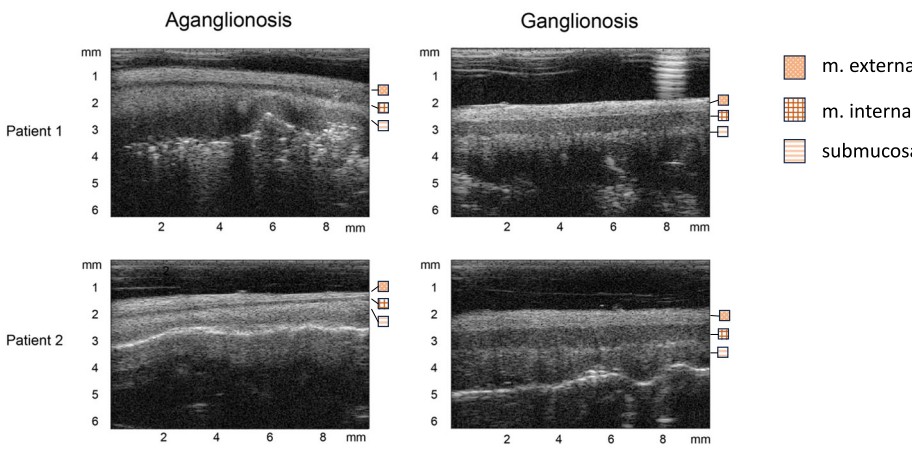

**Table 1 | Comparisons of thicknesses of the different histoanatomic layers (mm) and the ratio of muscularis interna/muscularis externa (numerical value)**

| Anatomical layer | Aganglionosis Median (range) | Ganglionosis Median (range) | *p*-value |
|---|---|---|---|
| Muscularis externa (mm) | 0.426 (0.196–0.685) | 0.466 (0.232–0.823) | 0.435 |
| Muscularis interna (mm) | 0.287 (0.147–0.441) | 0.562 (0.202–0.795) | <0.001 |
| Ratio interna/externa | 0.621 (0.309–1.616) | 1.225 (0.777–2.044) | <0.001 |

Paired Student's *t*-test. *p* < 0.05 was considered as significant.

be distributed normally. Differences between aganglionic and ganglionic bowel were analyzed using the Student's paired *t*-test. A *p*-value < 0.05 was considered as statistically significant. For inter-observer agreement, intra class correlation coefficient (ICC) was calculated[14].

### Ethical considerations

The study was approved by the Local Ethical Review Board at Lund University (DNR 2017/769) and the Swedish National Ethical Board (DNR 2023-01833.91). Consent from legal guardians was obtained before the study was performed, and all data were coded before the analyzes took place.

### Results

During the study period, 29 children were operated on for rectosigmoid Hirschsprung disease. Following the exclusion of cases not meeting the inclusion criteria due to total colonic aganglionosis and prior stoma (*n* = 6) or missing consent (*n* = 2), a total of 21 children were examined intra-operatively with UHF ultrasound. Of these 15 (71%) had images that, according to the histopathology reports, covered both aganglionic and ganglionic intestinal wall. Only these 15 examinations were included in the analyzes. For these 15 patients, the mean age at operation was 38 days (range 11–177 days) and mean weight was 4118 g (range 3100–7700 g). The mean length of the bowel in vivo was 22 cm (range 15–30 cm) before resection, 21 cm (range 14–26 cm) after resection but before formaldehyde fixation, and 18.5 cm (range 8.5–28.5 cm) after fixation.

The average waiting time from the collection of the intra-operative biopsy to the delivery of results by the pathologist was 44.5 min (range: 38–88 min). In four children (19%), an additional biopsy was needed as the first result was hypoganglionosis or aganglionosis. The repeated biopsy prolonged the time under anesthesia by a median of 39 min (range 30–63 min). All children had at least one biopsy taken, and in all cases, ganglion cells were present. No incorrect pathoanatomical diagnoses were given during the study. There were no intra-operative complications, but postoperatively, two children needed surgical intervention due to suspected abdominal wound dehiscence: one was confirmed, and one rejected.

In the analysis of the inter-observer agreement, the ICC value was 0.977, which is considered to be excellent. Measurements made by the

pediatric surgeon were chosen for analyzes since these were expected to reflect the proposed future clinical setting.

The thicknesses of the muscularis layers are presented in Table 1. The muscularis interna was significantly thicker in the ganglionic bowel in comparison to aganglionic. All but one of the included patients had a thicker muscularis interna in the ganglionic part of the bowel in comparison to the aganglionic (Fig. 2A). The ratio of the muscularis interna/externa was significantly greater in ganglionic bowel (0.621 vs 1.225, *p* < 0.001). All but two patients had a higher ratio in ganglionosis (Fig. 2B). The patient who did not present with a thicker muscularis interna in ganglionosis still had a greater ratio.

The echogenicity results of the different anatomical layers are presented in Table 2. In the submucosa, there was a significant difference with a higher echogenicity corresponding to a whiter submucosa in the aganglionic compared to the ganglionic part of the bowel (Fig. 3A). The echogenicity did not differ between aganglionic and ganglionic bowel in neither muscularis externa nor muscularis interna. The echogenicity ratios for both the muscularis externa/submucosa (Fig. 3B) and the muscularis interna/submucosa were significantly greater in aganglionic compared to the ganglionic bowel, synonymous with a more prominent whiteness difference between the compared layers in aganglionosis (Table 2).

### Discussion

This is a first report on the use of UHF ultrasound to determine the presence of ganglionic versus aganglionic bowel intra-operatively among children diagnosed with Hirschsprung disease. The study sought to determine whether measurable differences exist between aganglionic and ganglionic bowel, with patients serving as their own controls; consequently, no separate control group was included.

The results show that there was a significant measurable difference in the thickness of the muscularis interna between the aganglionic and ganglionic bowel. All but one of the included children had an increase in the muscularis interna from the aganglionic part to the ganglionic. In that child, the thickness of the muscularis interna in aganglionosis was the greatest among sections examined, and the difference was among the smallest (Fig. 2). The ratio of the thicknesses of the muscularis interna/externa also increased significantly in the ganglionic part of the bowel in comparison to

A.

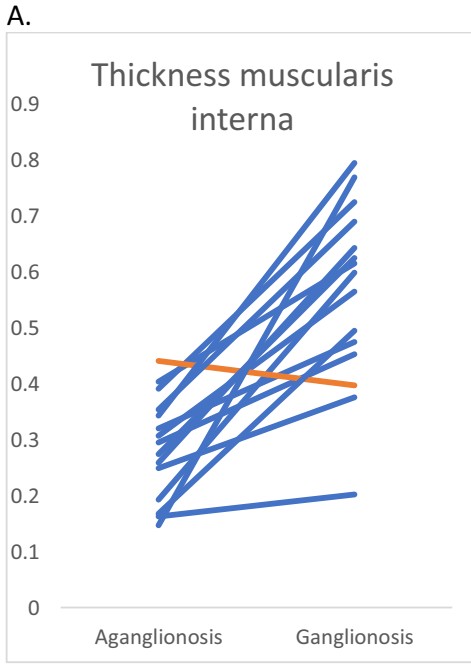

B.

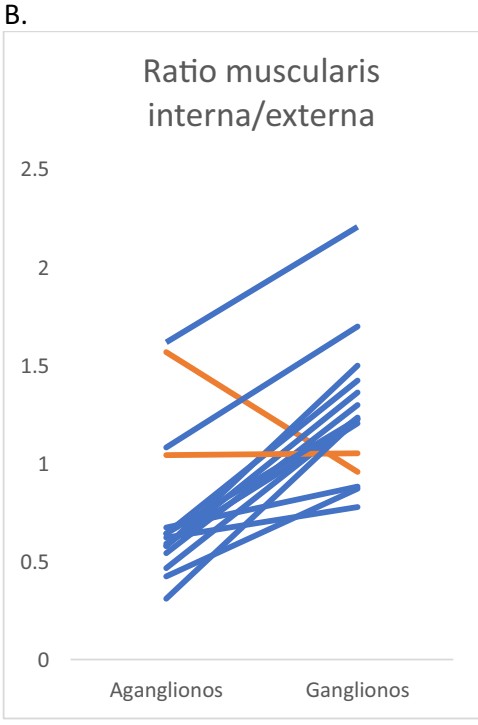

**Fig. 2 | Spaghetti plots of muscularis interna thickness and muscularis interna–to–muscularis externa ratio in Hirschsprung disease. A** shows individual trajectories of muscularis interna thickness (*y*-axis, mm) measured in aganglionic and ganglionated bowel using ultra-high-frequency ultrasound. **B** presents corresponding trajectories for the ratio of muscularis interna to muscularis externa (*y*-axis, unitless). Each line represents a single bowel segment, illustrating inter-segment variability within and between tissue types.

**Table 2 | Comparison of echogenicity of the histoanatomical layers in bowel wall and their ratios (numerical values)**

| Anatomical layer | Aganglionosis median (range) | Ganglionosis median (range) | *p*-value |
|---|---|---|---|
| Muscularis externa | 103 (23.3–126.9) | 104.1 (61.7–144.8) | 0.314 |
| Muscularis interna | 77 (21.3–96.5) | 73 (30.9–98.5) | 0.994 |
| Submucosa | 104.5 (45.2–123.6) | 81.7 (41.2–122.1) | 0.016 |
| Ratio interna/ externa | 0.73 (0.47–0.80) | 0.64 (0.40–0.80) | 0.060 |
| Ratioexterna/ submucosa | 1.03 (0.23–1.33) | 1.20 (0.80–1.87) | 0.003 |
| Ratiointerna/ submucosa | 0.72 (0.21–0.81) | 0.76 (0.62–0.92) | 0.033 |

Paired Student's *t*-test, $p < 0.05$ was considered as significant.

change. This observation may indicate that the muscularis interna is more sensitive to back-pressure.

Regarding echogenicity, there was also a significant difference, with a higher echogenicity in aganglionic submucosa. No differences in the echogenicity of the muscularis interna nor muscularis externa were found.

In a clinical perspective, the results of the study may prove to be significant. One of the most challenging parts during surgery for Hirschsprung disease is to know how much of the intestine needs to be resected. Although contrast enemas can be used preoperatively to estimate the aganglionic extension, intra-operative biopsies are still needed to determine ganglionosis, in order not to leave any aganglionic or transition zone[4]. The UHF ultrasound could play an important role in the decision of where to take the intra-operative biopsy and support the decision of on which level to resect the bowel. Such an instant method would shorten the time under anesthesia for the child, and probably even surgical workload and costs. In our study, we found that the child, on average, had an intra-operative waiting time of almost 45 min and additional for the patients who had to undergo a second biopsy. To shorten anesthesia time in children seems to be an urgent matter because studies have shown that a prolonged or repeated time under general anesthesia in early infancy might result in delayed neuropsychological development[15–18]. Another advantage might be the more accurate distinction of where the ganglionic part of the bowel starts, i.e., where the transition zone ends. The transition zone represents a particularly intriguing region, and our study group is currently collecting images from this part of the intestine. However, the material gathered thus far remains insufficient for analysis. Several studies have shown that the transition zone varies greatly from case to case and could be up to 22 cm. It has also been suggested that the surgeon should take the biopsy, or make the resection line, at least 5 cm orally from the suspected transition zone[19–21]. This margin is suggested to avoid mistakes and to resect any circumferentially irregular aganglionosis. This inexactness means that there is a risk that a longer segment of colon is resected than necessary, which can lead to difficulties in reaching down for anastomosis and possibly impacting functional outcome. The use of a UHF ultrasound replacing biopsies, avoiding time-consuming analyzes and personnel in the operating theater, might in addition lead to lower costs and workload. At our center, waiting times ranged from 28 to 88 min. This variability may be attributed to factors such as specimen transportation, freezing, sectioning, and preparation. In some cases, additional time was required when the pathologist sought a second opinion. Furthermore, if an endorectal transducer can be developed[22] it might mean that children do not need to undergo biopsies to confirm or reject the diagnosis, which in turn might spare families from anxiety while waiting for the diagnosis.

Overall, the results in this in vivo study highly confirmed what has been shown in our previous ex vivo studies on UHF ultrasound[10,11,13]. However, there were no differences regarding the echogenicity in the muscularis interna in the in vivo study, whereas in the ex vivo study, the same layer

the aganglionic, proving that there was not only a real but also relative increase of the muscularis interna in relation to the muscularis externa. The results are in line with ex vivo measurements performed prior to this study[10,12]. The muscularis interna was found to be thicker in the ganglionic segment, whereas the muscularis externa did not exhibit the same degree of

A

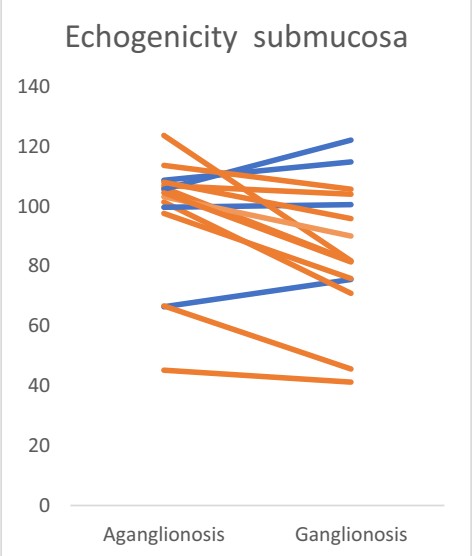

B

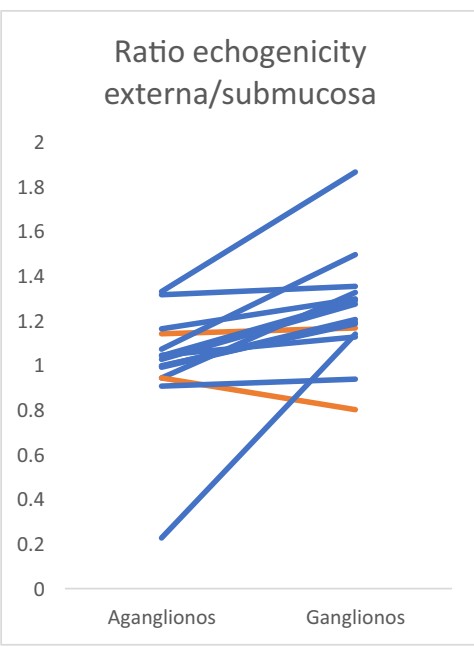

**Fig. 3 | Echogenicity profiles of aganglionic and ganglionated bowel wall in Hirschsprung disease. A** shows spaghetti plots illustrating variation in submucosal echogenicity in aganglionic and ganglionated bowel segments examined with ultra-high-frequency ultrasound. **B** depicts corresponding variation in the ratio of muscularis externa to submucosa. For both panels, the *y*-axis is unitless, and each line represents an individual bowel segment, highlighting within- and between-tissue variability.

presented with a significant higher echogenicity in aganglionic bowel[12]. Reasons for the differences in vivo compared to ex vivo are unclear, but speculatively devascularization may impact, causing other tissue composition features, which might be reflected in echogenicity variations. For clinical considerations, of course, the in vivo measurements attribute the most value.

The use of UHF ultrasound imaging has several advantages, e.g., it does not cause ionizing radiation or pain, nor does it require sedation, and it is associated with relatively low cost. But although the use of UHF ultrasound is feasible, it still presents certain limitations. For example, one disadvantage still is that the transducer is rather large, making the examination of small areas and compartments challenging. Due to the large size of the transducer, the shortest rectosigmoid extensions ending close to the peritoneal verge could not be reached, and therefore these patients could not be included. To examine lower regions, a small transducer, or even an endoscopic transducer, is warranted. Another limitation is that the image quality of UHF is very sensitive to movements, requiring full stillness during the scans. This was solved during the study by having one examiner working with the transducer and the other with the computer. Also, in line with traditional ultrasound, collecting images of high quality as well as interpretations of them are highly dependent on the ultrasound examiner. The development of a semi-automated computer program for interpreting the images will hopefully bridge this gap in the future.

Two strengths of the study were that the patients constituted their own controls and that the examinations were performed by trained examiners. The measures as compared between two observers, correlated very highly, and they were automatically calculated from multiple measurement points. A clear study limitation was the limited number of patients included, which might cause a type 2 error, explaining the absence of echogenicity differences. In the previous ex vivo study, where twice as many patients were included, more echogenicity differences were indicated. The size of the cohort for power is central, but the combination of studying a rare disease such as Hirschsprung disease and a new and unique technique like UHF ultrasound means a limited number of patients. Ideally, a definitive cut-off value for muscularis interna thickness would serve as a reliable criterion for identifying ganglionic bowel, as demonstrated in our ex vivo analyses[12]. At present, however, the in vivo dataset remains insufficient to establish such a threshold, and further data collection will be necessary before this can be defined. Another limitation is the fact that this is a single center study. To disseminate, the technique can be transferred to other conditions, or the machine can be transported to other centers. This technique has been described in pediatric conditions such as necrotizing enterocolitis and may also hold promise for intestinal diseases like Crohn's disease[9]. One consideration is that there was a spread of results between patients, especially regarding echogenicity, but also thicknesses showed a spread. Speculatively, this could be due to various sizes of patients, but in our previous studies, correlation analyzes of histoanatomy to age and weight have not shown homogenous results[13,23]. In the present study, we did not undertake any correlation analyzes to weight and age because the age group was already limited to patients 0–1 years old, with only small differences between patients. A larger cohort would be needed to be able to confirm or reject any associations. At the onset of our study, prior to the centralization of Hirschsprung disease and the establishment of standardized protocols, certain children were operated on at a notably early age. Remarkably, even in these early cases, the same alterations in the muscularis interna were observed. Future research would greatly benefit from establishing reference measurements of the various intestinal anatomical layers in children without intestinal disease, and from examining how these parameters correlate with age, weight, and specific anatomical landmarks within the intestine.

## Conclusion
The use of intra-operative UHF ultrasound shows great promise in the determination of ganglionic versus aganglionic bowel. Therefore, the UHF ultrasound method emerges as a strong candidate for enhancing the accuracy of intra-operative depiction of aganglionic extension and could contribute as a guide in the selection of intraoperative biopsy sites. Further studies are needed, as well as the development of an endoscopic UHF ultrasound transducer.

## Data availability
Access to the data is restricted to researchers affiliated with recognized academic institutions and is granted exclusively for non-commercial

research purposes. Access is conditional upon approval by an appropriate research ethics committee or equivalent regulatory body, where applicable. Requests for access to the data must be submitted to the corresponding author of this article via email. All requests will be assessed in accordance with applicable ethical and legal requirements, and applicants will be notified of the access decision within 30 days. Access to the data is subject to the execution of a legally binding Data Use Agreement. The data may be used solely for the specific research project approved under the agreement and must not be disclosed, transferred, or made available to any third party. The data will be provided in a de-identified format, and the recipient shall not attempt to re-identify any individual. Upon completion of the approved research project or review, all copies of the data must be permanently deleted in accordance with the terms of the Data Use Agreement.

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

## Acknowledgements
Ros Kenn, Medical Editor/Writer, who performed the language editing: http://roskenn.co.uk (accessed 18 June 2025). The Swedish Research Council Application and grant 2021-01569. Bengt Ihre Fellowship.

## Author contributions
C.G. and P.S. contributed with the study design. C.G., M.E., and P.S. contributed with the methodology. T.E., T.H., and M.E. contributed with the software. C.G. and M.E. contributed with the validation. C.G., M.E., and P.S. contributed with the formal analysis. C.G., K.H., L.T., and P.S. contributed with the investigation. C.G. and P.S. contributed with the resources. M.E. contributed with the data curation. C.G. and P.S. contributed with the original draft preparation. C.G., M.E., P.S., K.H., L.T., T.E., T.J., T.H., and M.C. contributed with the manuscript review and editing. C.G., P.S., and M.E. contributed with the visualization. P.S. and C.G. contributed with the project administration. C.G. was the Principal Investigator of the study. All authors have read and approved the final version of the manuscript.

## Funding

## Competing interests
The authors declare no competing interests.
