## [Transparent Peer Review file · Communications Medicine]

Intra-operative application of ultra-high frequency ultrasound facilitates differentiation of bowel wall characteristics between ganglionic and aganglionic segments during transanal endorectal pull-through for Hirschsprung disease

Corresponding Author: Dr Christina Graneli

Version 0:

Reviewer comments:

Reviewer #1

(Remarks to the Author)

This paper addresses the feasibility of a innovative method for intra operative diagnostic assessment of extent of innervative abnormalities in Hirschsprung patients undergoing reconstructive surgery. Although the paper is intriguing I have a number of concerns to be addressed

Introduction - this section sounds although I would shorten it to increase the readability of the paper

Methodology - There is the mandatory need to clarify some aspects namely the surgical procedure performed in these patients and the way USS was performed (did the operator perform circular or longitudinal assessment? These information are of utmost importance as intra operative USS is not feasible with laparoscopic assisted pull-through but with transanal but this means that real-time USS should be performed through the anus during dissection making the reliability of its use in real life definitely limited. Unless the operator can scan the entire colon during surgery I found it dangerous and limiting to do it either laparotomic or transanally.

Results - average age at surgery is too low. The authors should discuss their average age as since more than 5 years the entire scientific community agree that the ideal timing for surgery should be around 3-4 months of age. In this present series the average age at surgery is definitely too low. This aspect deserves clarification. It is not clear to me wether the authors can provide a clear cut-off value of interna muscularis propria layer to determine the normoganglionated bowel as this aspect would facilitate reproducibility of the results.

Discussion - the conclusions sound to me too much optimistic and over-reaching. The authors demonstrated a useful tool to help surgeons in addressing the thickness of colonic muscularis propria, which is directly related to the presence of a normal motility but I would be very very cautions in stating that this tool could replace intraoperative pathology. We would go back to old standards of care with visualizzation bowel thicnkess as a measure of normality. Of note, this approach led to a number of residual diseases that is unacceptable to date.

All in all the paper is nice but I wonder the usefulness of this technique to replace pathology. I would smoothen the discussion and conclusions and maybe I would speculate on the possobility to use this approach to lead the surgeons in performing only the useful biopsie e reducing timing of pathology assessment. I would never rely on USS instead of pathology to perform an anastomosis as we are watching the results on motility and not innervation per sé.

This paper deserves major revisions but even so I am a bit worried about the message delivered by the authors.

Reviewer #2

(Remarks to the Author)

Thank you for the opportunity to review this manuscript.

The authors present a new method for the intraoperative investigation of the large bowel in patients with Hirschsprung

disease. It is crucial for the correct surgical resection to discriminate the normoganglionic part in the colon from the aganglionic part. The Goldstandard is to obtain and evaluate intraoperative biopsies.

The authors here present the use of intraoperative ultrasound to distinguish between ganglionic and aganglionic bowel. Basically this is a novel approach, but the title is actually misleading as the ultrasound can not display the ganglion cells of the bowel wall. This method actually measures the muscle wall thickness and extrapolates to the possible presence of ganglion cells after having the histology of a biopsy.

The inclusion and exclusion criteria are clear, there is an appropriate ethical consent.

The methods section requires some clarification, as it only states that parts of the suspected aganglionic (sigmoid) and parts of the suspected ganglionic (descendens) bowel have been investigated with ultrasound. It is now very well known, that circular (inner) muscle layer is uniformly developed but not the longitudinal (outer) muscle layer is mainly expressed in the taenia. This means, that the circular muscle is usually thicker than the longitudinal muscle in non taenia areas, whereas the longitudinal is thicker in taenia areas. How did the authors make sure to always investigating in comparable regions? What are the normal variants of the thickness of inner and outer muscle layer of the colon in children?

It is usually expected that muscle layers proximal to a stenosis or contraction (as in Hirschsprung disease) will be hypertrophied? How do the authors explain the opposite findings in their study.

The number of included patients is rather small. There is no control group, where the investigators did not intraoperative biopsies and proceeded straight to the resection in accordance to the US results and would have done a postoperative confirmation of the presence of normal ganglion cells on the proximal resection margin. With this method the time of surgery is not shortened and I am in doubt, that ultrasound will really replace intraoperative biopsies.

Reviewer #3

(Remarks to the Author)

Dear Authors,

This research paper addresses a condition that represents one of the most relevant conditions in the field of pediatric surgery. Although considerable evidence is currently available on this disease, it may still pose a challenge for physicians, since the clinical presentation could be misdiagnosed as other conditions, and because there is no consensus in the literature regarding its management and the most appropriate surgical technique. The study focuses on the diagnosis of Hirschsprung's disease and on the ultrasound parameters used to distinguish ganglionic from aganglionic bowel. I provide below some considerations to help improve the manuscript:

- The manuscript refers to patients treated with the transanal endorectal pull-through (TERPT) technique. I suggest clarifying this aspect in the title and in the abstract.

- Which form of Hirschsprung's disease affected the patients included in the study? Please specify this information in the text.

- The paper refers to the use of intraoperative ultrasound to differentiate ganglionic from aganglionic bowel during surgical correction. I recommend highlighting this aspect in the "hypothesis" section of the abstract.

- Line 54: In your centre, do you have experience in obtaining the biopsy specimen from an area more proximal than the transition zone to avoid performing multiple biopsies?

If so, at what distance from the transition zone are these biopsies usually taken?

- From 2020 to 2024, were all patients with Hirschsprung's disease treated with the same surgical technique? If some patients underwent different surgical procedures, please specify this information in the exclusion criteria.

- Both a surgeon and an engineer performed the measurements. Who performed the ultrasound examinations? Was it a surgeon or a radiologist?

- The waiting time reported for biopsy results ranged from 38 to 88 minutes. According to the authors, what accounts for such a wide variability in timing?

- Do the authors consider performing measurements of the intestinal wall, analysing the difference between the ganglionic bowel, the aganglionic bowel and the transition zone?

- I recommend integrating the discussion section with data from the literature on the use of UHF ultrasound, including its possible applications in other pediatric surgical conditions.

- Do the authors confirm the hypothesis that, despite its limitations, UHF ultrasound could replace frozen section biopsy in distinguishing aganglionic from ganglionic colon?

- I suggest including a paragraph on the limitations of the study (e.g., single-centre study, small sample size).

- The quality of the English language is fine, but I recommend further revision to improve the fluency and academic style of the text (e.g., line 50: "targeted" instead of "aimed").

The topic of the manuscript about the application of UHF ultrasound on intestinal wall examination to differentiate ganglionic from aganglionic bowel could be significant. However, the manuscript could benefit from a major revision before being considered suitable for publication.

Reviewer #4

(Remarks to the Author)

I co-reviewed this manuscript with one of the reviewers who provided the listed reports. This is part of the Communications Medicine initiative to facilitate training in peer review and to provide appropriate recognition for Early Career Researchers

who co-review manuscripts.

Version 1:

Reviewer comments:

Reviewer #1

(Remarks to the Author)

The Authors addressed present reviewer's concerns and improved the paper significantly. Based on these consideration, in its present form the paper is now suitable for publication

Reviewer #3

(Remarks to the Author)

I have carefully reviewed the revised version of the manuscript.

The authors have adequately addressed all the previously raised comments and concerns. The revisions have improved the clarity and overall quality of the manuscript.

I therefore consider the manuscript suitable for publication in its current form.

Reviewer #4

(Remarks to the Author)

I co-reviewed this manuscript with one of the reviewers who provided the listed reports. This is part of the Communications Medicine initiative to facilitate training in peer review and to provide appropriate recognition for Early Career Researchers who co-review manuscripts.

Reviewer 1	Author comment	Manuscript change
There is the mandatory need to clarify some aspects namely the surgical procedure performed in these patients and the way USS was performed (did the operator perform circular or longitudinal assessment?)	All patients underwent a transanal endorectal pull-through procedure, with assistance of a subumbilical mini-laparotomy. A muscular cuff was constructed approximately 2 cm proximal to the dentate line, in accordance with a modified Soave technique. Ultra-high-frequency ultrasonographic imaging was acquired from the serosal aspect of the intestine, utilizing longitudinal orientations	Added to the Method: All patients underwent a transanal endorectal pull-through procedure, with assistance of a subumbilical mini-laparotomy. A muscular cuff was constructed approximately 2 cm proximal to the dentate line, in accordance with a modified Soave technique. Ultra-high-frequency ultrasonographic imaging was acquired from the serosal aspect of the intestine, utilizing longitudinal orientations
Average age at surgery is too low. The authors should discuss their average age as since more than 5 years the entire scientific community agree that the ideal timing for surgery should be around 3-4 months of age. In this present series the average age at surgery is definitely too low. This aspect deserves clarification.	Thank you for this observation; it is indeed accurate for this study. This can be explained by the fact that two of the patients underwent surgery prior to the centralization of Hirschsprung disease management in our country, at a time when standardized protocols were not yet established. Interestingly, these patients were operated on at a very early stage, and despite this, we observed the same differences in the muscularis interna. We have incorporated this point into the discussion section.	Added to the Discussion: At the onset of our study, prior to the centralization of Hirschsprung disease and the establishment of standardized protocols, certain children were operated on at a notably early age. Remarkably, even in these early cases, the same alterations in the muscularis interna were observed.
It is not clear to me whether the authors can provide a clear cut-off value of interna muscularis propria layer to determine the normoganglionated bowel as this aspect would facilitate reproducibility of the results.	Thank you for raising this question. Ideally, it would be preferable to establish a clear cut-off value for the thickness of the muscularis interna as a criterion for identifying ganglionic bowel, as we have done in our ex vivo analyses (Tebin's reference). However, at this stage, we consider the in vivo dataset too limited to define such a numerical	Added to the discussion: Ideally, a definitive cut-off value for muscularis interna thickness would serve as a reliable criterion for identifying ganglionic bowel, as demonstrated in our ex vivo analyses (Tebin's reference). At present, however, the in vivo dataset remains insufficient to establish such a threshold,

	threshold. Additional data collection will be required before this can be determined.	and further data collection will be necessary before this can be defined.
Discussion - the conclusions sound to me too much optimistic and over-reaching. The authors demonstrated a useful tool to help surgeons in addressing the thickness of colonic muscularis propria, which is directly related to the presence of a normal motility but I would be very very cautions in stating that this tool could replace intraoperative pathology. We would go back to old standards of care with visualizzation bowel thickness as a measure of normality. Of note, this approach led to a number of residual diseases that is unacceptable to date. All in all the paper is nice but I wonder the usefulness of this technique to replace pathology. I would smoothen the discussion and conclusions and maybe I would speculate on the possobility to use this approach to lead the surgeons in performing only the useful biopsie e reducing timing of pathology assessment. I would never rely on USS instead of pathology to perform an anastomosis as we are watching the results on motility and not innervation per sé.	We appreciate this observation and agree that a more cautious tone is warranted. At present, UHF ultrasonography cannot replace histological assessment. In the discussion, we propose that UHF imaging may have potential utility only in guiding the selection of intraoperative biopsy sites. Furthermore, we have clarified that, in the future, UHF ultrasonography could contribute to determining the optimal level of bowel resection.	Added to the Conclusion: Therefore, the UHF ultrasound method emerges as a strong candidate for enhancing the accuracy of intra-operative depiction of aganglionic extension and could contribute as a guide in the selection of intraoperative biopsy sites.
Reviewer 2		
Basically this is a novel approach, but the title is actually misleading as the ultrasound can not display the ganglion cells of the bowel wall.	We understand your point and would like to propose a clearer title:	Title changed: Intraoperative application of ultra-high-frequency ultrasound facilitates differentiation of bowel wall

	Intraoperative application of ultra-high-frequency ultrasound facilitates differentiation of bowel wall characteristics between ganglionic and aganglionic segments during transanal endorectal pull-through for Hirschsprung disease.	characteristics between ganglionic and aganglionic segments during transanal endorectal pull-through for Hirschsprung disease.
The methods section requires some clarification, as it only states that parts of the suspected aganglionic (sigmoid) and parts of the suspected ganglionic (descendens) bowel have been investigated with ultrasound. It is now very well known, that circular (inner) muscle layer is uniformly developed but not the longitudinal (outer) muscle layer is mainly expressed in the taeni This means, that the circular muscle is usually thicker than the longitudinal muscle in non taenia areas, whereas the longitudinal is thicker in taenia areas. How did the authors make sure to always investigating in comparable regions?	Thank you for the opportunity to improve and clarify, and for this very insightful question. Images were obtained from the anti-mesenteric side of the colonic wall, specifically over the taenia, to ensure a consistent anatomical landmark and allow all measurements to be performed as uniformly as possible.	Added to the Method: Images were obtained from the anti-mesenteric side of the colonic wall, specifically over the taenia, to ensure a consistent anatomical landmark and allow all measurements to be performed as uniformly as possible.
What are the normal variants of the thickness of inner and outer muscle layer of the colon in children?	Unfortunately, to the best of our knowledge, no studies have investigated this to date. It would be highly valuable for future research to establish these measurements in children without intestinal disease and to examine how they correlate with age, weight, and various anatomical landmarks within the intestine. We have added the following point to the discussion:	Added to the Discussion: Future research would greatly benefit from establishing reference measurements of the various intestinal anatomical layers in children without intestinal disease, and from examining how these parameters correlate with age, weight, and specific anatomical landmarks within the intestine.
It is usually expected that muscle layers proximal to a stenosis or contraction (as in Hirschsprung disease) will be hypertrophied? How do	Thank you for pointing this out. We observed that the muscularis interna was thicker (hypertrophied?) in the ganglionic segment.	Added to the discussion: The muscularis interna was found to be thicker in the ganglionic segment,

the authors explain the opposite findings in their study.	Conversely, this was not the case for the muscularis externa to the same extent, which may suggest that the muscularis interna is more responsive to back-pressure	whereas the muscularis externa did not exhibit the same degree of change. This observation may indicate that the muscularis interna is more sensitive to back-pressure.
The number of included patients is rather small. There is no control group, where the investigators did nit intraoperative biopsies and proceeded straight to the resection in accordance to the US results and would have done an postoperative confirmation of the presence of normal ganglion cells on the proximal resection margin.	Thank you for the comment. It is correct that we did not include a control group, as this was an initial study aimed at determining whether measurable differences exist between aganglionic and ganglionic bowel. As you suggest, such an approach could represent an important step. We have added this point to the discussion.	Added to the discussion: The study sought to determine whether measurable differences exist between aganglionic and ganglionic bowel, with patients serving as their own controls; consequently, no separate control group was included.
Reviewer 3		
The manuscript refers to patients treated with the transanal endorectal pull-through (TERPT) technique. I suggest clarifying this aspect in the title and in the abstract.	We understand a need in title for clarifying that the ultrasound was performed intraoperatively, during the transanal endorectal pull-through procedure. Intraoperative application of ultra-high-frequency ultrasound facilitates differentiation of bowel wall characteristics between ganglionic and aganglionic segments during transanal endorectal pull-through for Hirschsprung disease.	Title changed: Intraoperative application of ultra-high-frequency ultrasound facilitates differentiation of bowel wall characteristics between ganglionic and aganglionic segments during transanal endorectal pull-through for Hirschsprung disease.
Which form of Hirschsprung's disease affected the patients included in the study? Please specify this information in the text.	The only form of Hirschsprung disease included was rectosigmoid. This is stated in the Aim, Methods section as well as the Aim in the abstract.	The aim of this study was to determine UHF ultrasound's effectiveness in distinguishing ganglionic from aganglionic bowel wall during trans-anal endo-rectal pull-through (TERPT) procedure for rectosigmoid Hirschsprung disease.

The paper refers to the use of intraoperative ultrasound to differentiate ganglionic from aganglionic bowel during surgical correction. I recommend highlighting this aspect in the “hypothesis” section of the abstract	The hypothesis was that intraoperative ultrasound could differentiate ganglionic from aganglionic bowel during surgical correction	Added to the Abstract: Hypothesis: ultra-high frequency (UHF) ultrasound may distinguish ganglionic from aganglionic bowel during surgery for Hirschsprung disease.
Line 54: In your centre, do you have experience in obtaining the biopsy specimen from an area more proximal than the transition zone to avoid performing multiple biopsies? If so, at what distance from the transition zone are these biopsies usually taken?	Thank you for this question. We aim to obtain the biopsies some centimetres but maximum approximately 5 cm above the suspected transition zone. We have added this clarification to the Methods section.	Added to the Method: Inclusion criteria were rectosigmoid aganglionosis operated on with TERPT and assistance of a subumbilical mini-laparotomy , age 0-12 months and no preoperative stoma. Intraoperative biopsies were collected approximately 5 cm proximal to the suspected transition zone.
From 2020 to 2024, were all patients with Hirschsprung’s disease treated with the same surgical technique? If some patients underwent different surgical procedures, please specify this information in the exclusion criteria	Thank you for highlighting this point. The same standardized surgical technique was applied to all patients diagnosed with recto-sigmoid Hirschsprung disease. This has now been specified in the Methods section for clarity	Added to the Method: Included patients underwent a TERPT procedure with assistance of a subumbilical mini-laparotomy. A muscular cuff was constructed approximately 2 cm proximal to the dentate line, in accordance with a modified Soave technique. Exclusion criteria included aganglionosis extending beyond the recto-sigmoid, lack of consent, and laparoscopic procedures.
Both a surgeon and an engineer performed the measurements. Who performed the ultrasound examinations? Was it a surgeon or a radiologist?	All ultrasound examinations were performed clinically by the same three pediatric surgeons. The engineer was only involved in reviewing the acquired images and assisting with the adjustment of ultrasound settings. This clarification has been added to the Methods section	Added to the Method: All ultrasound examinations were performed clinically by the same three pediatric surgeons.
The waiting time reported for biopsy results ranged from 38 to 88 minutes. According	We agree that the waiting time varies considerably. While we can only speculate	Added to the Methode:

to the authors, what accounts for such a wide variability in timing?	on individual causes, delays may occur due to specimen transport, freezing and sectioning, or preparation. Additional time is sometimes required if the pathologist seeks a second opinion or during sample processing. The procedure followed in our department has been clarified in the Methods section, and potential reasons to delay in Discussion.	The biopsies were subsequently transferred to the pathology department for freezing, sectioning, and preparation. All specimens were evaluated by pediatric pathologists. Discussion: At our center, waiting times ranged from 28 to 88 minutes. This variability may be attributed to factors such as specimen transportation, freezing, sectioning, and preparation. In some cases, additional time was required when the pathologist sought a second opinion.
Do the authors consider performing measurements of the intestinal wall, analysing the difference between the ganglionic bowel, the aganglionic bowel and the transition zone?	We agree that this is a highly interesting aspect of the enigma of Hirschsprung disease. At present, we are still collecting images of the transition zone; however, the material gathered so far is insufficient for publication	Added to the Discussion: The transition zone represents a particularly intriguing region, and our study group is currently collecting images from this part of the intestine. However, the material gathered thus far remains insufficient for analysis.
Do the authors confirm the hypothesis that, despite its limitations, UHF ultrasound could replace frozen section biopsy in distinguishing aganglionic from ganglionic colon?	We appreciate this observation and would like to include a more cautious tone. At present, UHF ultrasonography cannot replace histological assessment. In the discussion, we propose that UHF imaging may have potential utility in guiding the selection of intraoperative biopsy sites. Furthermore, we have clarified that, in the future, UHF ultrasonography could contribute to determining the optimal level of bowel resection	Added to the Conclusion Therefore, the UHF ultrasound method emerges as a strong candidate for enhancing the accuracy of intra-operative depiction of aganglionic extension and could contribute as a guide in the selection of intraoperative biopsy sites
I recommend integrating the discussion section with data from the literature on the use of UHF ultrasound, including its possible applications in other pediatric surgical conditions.	Thank you for the suggestion. We have now added that we see potential, for example, in delineating healthy and affected bowel during pediatric surgery for other conditions, such as	Added to the Discussion: This technique has been described in pediatric conditions such as necrotizing enterocolitis and may also hold promise for

	necrotizing enterocolitis or Crohn's disease.	intestinal diseases like Crohn's disease.
I suggest including a paragraph on the limitations of the study (e.g., single-centre study, small sample size)	We agree that it is important to acknowledge the limitations. In the Discussion section, we have added the single-center limitation to the paragraph addressing the restricted material.	Added to the Discussion: Another limitation is the fact that this is a single center study Since before in the Discussion: A clear study limitation was the limited number of patients included which might cause a type 2 error, explaining the absence of echogenicity differences.

Reviewer comments 260212:

Reviewer 1	Author comment	Manuscript change
The Authors addressed present reviewer's concerns and improved the paper significantly. Based on this consideration, in its present form the paper is now suitable for publication	Thank you for reviewing our manuscript and providing valuable insights and questions that helped us improve it	
Reviewer 2		
(Remarks to the Author): Assessed by Editor		
Reviewer 3		
I have carefully reviewed the revised version of the manuscript. The authors have adequately addressed all the previously raised comments and concerns. The revisions have improved the clarity and overall quality of the manuscript. I therefore consider the manuscript suitable for publication in its current form.	Thank you for your comments and questions regarding the manuscript, which have helped us improve it during this process.	
Reviewer 4		
I co-reviewed this manuscript with one of the reviewers who provided the listed reports. This is part of the Communications Medicine initiative to facilitate training in peer review and to provide appropriate recognition for Early Career Researchers who co-review manuscripts.		

Reviewer comments 250917:

Reviewer 1	Author comment	Manuscript change
There is the mandatory need to clarify some aspects namely the surgical procedure performed in these patients and the way USS was performed (did the operator perform circular or longitudinal assessment?)	All patients underwent a transanal endorectal pull-through procedure, with assistance of a subumbilical mini-laparotomy. A muscular cuff was constructed approximately 2 cm proximal to the dentate line, in accordance with a modified Soave technique. Ultra-high-frequency ultrasonographic imaging was acquired from the serosal aspect of the intestine, utilizing longitudinal orientations	Added to the Method: All patients underwent a transanal endorectal pull-through procedure, with assistance of a subumbilical mini-laparotomy. A muscular cuff was constructed approximately 2 cm proximal to the dentate line, in accordance with a modified Soave technique. Ultra-high-frequency ultrasonographic imaging was acquired from the serosal aspect of the intestine, utilizing longitudinal orientations
Average age at surgery is too low. The authors should discuss their average age as since more than 5 years the entire scientific community agree that the ideal timing for surgery should be around 3-4 months of age. In this present series the average age at surgery is definitely too low. This aspect deserves clarification.	Thank you for this observation; it is indeed accurate for this study. This can be explained by the fact that two of the patients underwent surgery prior to the centralization of Hirschsprung disease management in our country, at a time when standardized protocols were not yet established. Interestingly, these patients were operated on at a very early stage, and despite this, we observed the same differences in the muscularis interna. We have incorporated this point into the discussion section.	Added to the Discussion: At the onset of our study, prior to the centralization of Hirschsprung disease and the establishment of standardized protocols, certain children were operated on at a notably early age. Remarkably, even in these early cases, the same alterations in the muscularis interna were observed.
It is not clear to me whether the authors can provide a clear cut-off value of interna muscularis propria layer to determine the normoganglionated bowel as this aspect would facilitate reproducibility of the results.	Thank you for raising this question. Ideally, it would be preferable to establish a clear cut-off value for the thickness of the muscularis interna as a criterion for identifying ganglionic bowel, as we have done in our ex vivo analyses (Tebin's reference). However, at this stage, we consider the in	Added to the discussion: Ideally, a definitive cut-off value for muscularis interna thickness would serve as a reliable criterion for identifying ganglionic bowel, as demonstrated in our ex vivo analyses (Tebin's reference). At present, however, the in vivo dataset

	vivo dataset too limited to define such a numerical threshold. Additional data collection will be required before this can be determined.	remains insufficient to establish such a threshold, and further data collection will be necessary before this can be defined.
Discussion - the conclusions sound to me too much optimistic and over-reaching. The authors demonstrated a useful tool to help surgeons in addressing the thickness of colonic muscularis propria, which is directly related to the presence of a normal motility but I would be very very cautions in stating that this tool could replace intraoperative pathology. We would go back to old standards of care with visualizzation bowel thickness as a measure of normality. Of note, this approach led to a number of residual diseases that is unacceptable to date. All in all the paper is nice but I wonder the usefulness of this technique to replace pathology. I would smoothen the discussion and conclusions and maybe I would speculate on the possobility to use this approach to lead the surgeons in performing only the useful biopsie e reducing timing of pathology assessment. I would never rely on USS instead of pathology to perform an anastomosis as we are watching the results on motility and not innervation per sé.	We appreciate this observation and agree that a more cautious tone is warranted. At present, UHF ultrasonography cannot replace histological assessment. In the discussion, we propose that UHF imaging may have potential utility only in guiding the selection of intraoperative biopsy sites. Furthermore, we have clarified that, in the future, UHF ultrasonography could contribute to determining the optimal level of bowel resection.	Added to the Conclusion: Therefore, the UHF ultrasound method emerges as a strong candidate for enhancing the accuracy of intra-operative depiction of aganglionic extension and could contribute as a guide in the selection of intraoperative biopsy sites.
Reviewer 2		
Basically this is a novel approach, but the title is actually misleading as the ultrasound can not display	We understand your point and would like to propose a clearer title:	Title changed: Intraoperative application of ultra-high-frequency

the ganglion cells of the bowel wall.	Intraoperative application of ultra-high-frequency ultrasound facilitates differentiation of bowel wall characteristics between ganglionic and aganglionic segments during transanal endorectal pull-through for Hirschsprung disease.	ultrasound facilitates differentiation of bowel wall characteristics between ganglionic and aganglionic segments during transanal endorectal pull-through for Hirschsprung disease.
The methods section requires some clarification, as it only states that parts of the suspected aganglionic (sigmoid) and parts of the suspected ganglionic (descendens) bowel have been investigated with ultrasound. It is now very well known, that circular (inner) muscle layer is uniformly developed but not the longitudinal (outer) muscle layer is mainly expressed in the taeni This means, that the circular muscle is usually thicker than the longitudinal muscle in non taenia areas, whereas the longitudinal is thicker in taenia areas. How did the authors make sure to always investigating in comparable regions?	Thank you for the opportunity to improve and clarify, and for this very insightful question. Images were obtained from the anti-mesenteric side of the colonic wall, specifically over the taenia, to ensure a consistent anatomical landmark and allow all measurements to be performed as uniformly as possible.	Added to the Method: Images were obtained from the anti-mesenteric side of the colonic wall, specifically over the taenia, to ensure a consistent anatomical landmark and allow all measurements to be performed as uniformly as possible.
What are the normal variants of the thickness of inner and outer muscle layer of the colon in children?	Unfortunately, to the best of our knowledge, no studies have investigated this to date. It would be highly valuable for future research to establish these measurements in children without intestinal disease and to examine how they correlate with age, weight, and various anatomical landmarks within the intestine. We have added the following point to the discussion:	Added to the Discussion: Future research would greatly benefit from establishing reference measurements of the various intestinal anatomical layers in children without intestinal disease, and from examining how these parameters correlate with age, weight, and specific anatomical landmarks within the intestine.
It is usually expected that muscle layers proximal to a stenosis or contraction (as in Hirschsprung disease) will be hypertrophied? How do	Thank you for pointing this out. We observed that the muscularis interna was thicker (hypertrophied?) in the ganglionic segment.	Added to the discussion: The muscularis interna was found to be thicker in the ganglionic segment,

the authors explain the opposite findings in their study.	Conversely, this was not the case for the muscularis externa to the same extent, which may suggest that the muscularis interna is more responsive to back-pressure	whereas the muscularis externa did not exhibit the same degree of change. This observation may indicate that the muscularis interna is more sensitive to back-pressure.
The number of included patients is rather small. There is no control group, where the investigators did nit intraoperative biopsies and proceeded straight to the resection in accordance to the US results and would have done an postoperative confirmation of the presence of normal ganglion cells on the proximal resection margin.	Thank you for the comment. It is correct that we did not include a control group, as this was an initial study aimed at determining whether measurable differences exist between aganglionic and ganglionic bowel. As you suggest, such an approach could represent an important step. We have added this point to the discussion.	Added to the discussion: The study sought to determine whether measurable differences exist between aganglionic and ganglionic bowel, with patients serving as their own controls; consequently, no separate control group was included.
Reviewer 3		
The manuscript refers to patients treated with the transanal endorectal pull-through (TERPT) technique. I suggest clarifying this aspect in the title and in the abstract.	We understand a need in title for clarifying that the ultrasound was performed intraoperatively, during the transanal endorectal pull-through procedure. Intraoperative application of ultra-high-frequency ultrasound facilitates differentiation of bowel wall characteristics between ganglionic and aganglionic segments during transanal endorectal pull-through for Hirschsprung disease.	Title changed: Intraoperative application of ultra-high-frequency ultrasound facilitates differentiation of bowel wall characteristics between ganglionic and aganglionic segments during transanal endorectal pull-through for Hirschsprung disease.
Which form of Hirschsprung's disease affected the patients included in the study? Please specify this information in the text.	The only form of Hirschsprung disease included was rectosigmoid. This is stated in the Aim, Methods section as well as the Aim in the abstract.	The aim of this study was to determine UHF ultrasound's effectiveness in distinguishing ganglionic from aganglionic bowel wall during trans-anal endo-rectal pull-through (TERPT) procedure for rectosigmoid Hirschsprung disease.

The paper refers to the use of intraoperative ultrasound to differentiate ganglionic from aganglionic bowel during surgical correction. I recommend highlighting this aspect in the “hypothesis” section of the abstract	The hypothesis was that intraoperative ultrasound could differentiate ganglionic from aganglionic bowel during surgical correction	Added to the Abstract: Hypothesis: ultra-high frequency (UHF) ultrasound may distinguish ganglionic from aganglionic bowel during surgery for Hirschsprung disease.
Line 54: In your centre, do you have experience in obtaining the biopsy specimen from an area more proximal than the transition zone to avoid performing multiple biopsies? If so, at what distance from the transition zone are these biopsies usually taken?	Thank you for this question. We aim to obtain the biopsies some centimetres but maximum approximately 5 cm above the suspected transition zone. We have added this clarification to the Methods section.	Added to the Method: Inclusion criteria were rectosigmoid aganglionosis operated on with TERPT and assistance of a subumbilical mini-laparotomy , age 0-12 months and no preoperative stoma. Intraoperative biopsies were collected approximately 5 cm proximal to the suspected transition zone.
From 2020 to 2024, were all patients with Hirschsprung’s disease treated with the same surgical technique? If some patients underwent different surgical procedures, please specify this information in the exclusion criteria	Thank you for highlighting this point. The same standardized surgical technique was applied to all patients diagnosed with recto-sigmoid Hirschsprung disease. This has now been specified in the Methods section for clarity	Added to the Method: Included patients underwent a TERPT procedure with assistance of a subumbilical mini-laparotomy. A muscular cuff was constructed approximately 2 cm proximal to the dentate line, in accordance with a modified Soave technique. Exclusion criteria included aganglionosis extending beyond the recto-sigmoid, lack of consent, and laparoscopic procedures.
Both a surgeon and an engineer performed the measurements. Who performed the ultrasound examinations? Was it a surgeon or a radiologist?	All ultrasound examinations were performed clinically by the same three pediatric surgeons. The engineer was only involved in reviewing the acquired images and assisting with the adjustment of ultrasound settings. This clarification has been added to the Methods section	Added to the Method: All ultrasound examinations were performed clinically by the same three pediatric surgeons.
The waiting time reported for biopsy results ranged from 38 to 88 minutes. According	We agree that the waiting time varies considerably. While we can only speculate	Added to the Methode:

to the authors, what accounts for such a wide variability in timing?	on individual causes, delays may occur due to specimen transport, freezing and sectioning, or preparation. Additional time is sometimes required if the pathologist seeks a second opinion or during sample processing. The procedure followed in our department has been clarified in the Methods section, and potential reasons to delay in Discussion.	The biopsies were subsequently transferred to the pathology department for freezing, sectioning, and preparation. All specimens were evaluated by pediatric pathologists. Discussion: At our center, waiting times ranged from 28 to 88 minutes. This variability may be attributed to factors such as specimen transportation, freezing, sectioning, and preparation. In some cases, additional time was required when the pathologist sought a second opinion.
Do the authors consider performing measurements of the intestinal wall, analysing the difference between the ganglionic bowel, the aganglionic bowel and the transition zone?	We agree that this is a highly interesting aspect of the enigma of Hirschsprung disease. At present, we are still collecting images of the transition zone; however, the material gathered so far is insufficient for publication	Added to the Discussion: The transition zone represents a particularly intriguing region, and our study group is currently collecting images from this part of the intestine. However, the material gathered thus far remains insufficient for analysis.
Do the authors confirm the hypothesis that, despite its limitations, UHF ultrasound could replace frozen section biopsy in distinguishing aganglionic from ganglionic colon?	We appreciate this observation and would like to include a more cautious tone. At present, UHF ultrasonography cannot replace histological assessment. In the discussion, we propose that UHF imaging may have potential utility in guiding the selection of intraoperative biopsy sites. Furthermore, we have clarified that, in the future, UHF ultrasonography could contribute to determining the optimal level of bowel resection	Added to the Conclusion Therefore, the UHF ultrasound method emerges as a strong candidate for enhancing the accuracy of intra-operative depiction of aganglionic extension and could contribute as a guide in the selection of intraoperative biopsy sites
I recommend integrating the discussion section with data from the literature on the use of UHF ultrasound, including its possible applications in other pediatric surgical conditions.	Thank you for the suggestion. We have now added that we see potential, for example, in delineating healthy and affected bowel during pediatric surgery for other conditions, such as	Added to the Discussion: This technique has been described in pediatric conditions such as necrotizing enterocolitis and may also hold promise for

	necrotizing enterocolitis or Crohn's disease.	intestinal diseases like Crohn's disease.
I suggest including a paragraph on the limitations of the study (e.g., single-centre study, small sample size)	We agree that it is important to acknowledge the limitations. In the Discussion section, we have added the single-center limitation to the paragraph addressing the restricted material.	Added to the Discussion: Another limitation is the fact that this is a single center study Since before in the Discussion: A clear study limitation was the limited number of patients included which might cause a type 2 error, explaining the absence of echogenicity differences.